**Data Availability Statement:** All relevant data are within the paper and its Supporting Information files.

# Targeted metabolomics investigation of metabolic markers of *Mycobacterium tuberculosis* in the cerebrospinal fluid of paediatric patients with tuberculous meningitis

**Victory Samuel[1], Regan Solomons[2], Shayne Mason[1]***

**1** Biochemistry Department, Focus Area for Human Metabolomics, North-West University, Potchefstroom, South Africa, **2** Faculty of Medicine and Health Sciences, Department of Paediatrics and Child Health, Stellenbosch University, Cape Town, South Africa

* nmr.nwu@gmail.com

## Abstract

### Objective

To investigate metabolic markers linked to *Mycobacterium tuberculosis* (*M. tb*) in the cerebrospinal fluid (CSF) of a South African cohort of paediatric tuberculous meningitis (TBM).

### Methods

Targeted proton magnetic resonance ($^1$H-NMR) spectroscopy and two-dimensional gas chromatography coupled with time-of-flight mass spectrometry (GCxGC-TOFMS) metabolomics approaches were used to evaluate *M. tb*-linked metabolites in the CSF of 21 definite cases of TBM and 25 control cases. Uni- and multivariate statistical analyses were performed.

### Results

Four statistically significant metabolites were identified to discriminate TBM cases from controls. Mannose and arabinose were found at lower concentrations in the TBM group. Nonanoic acid and propanoic acid were found in higher concentrations in the definite TBM group.

### Conclusions

We identified the novel presence of nonanoic acid for the first time as a *M. tb*-linked marker in the CSF of cases of TBM, possibly as a degradation product of the *M. tb* cell wall. Propanoic acid can be related to perturbed brain neuro-energetics and neuro-inflammation in TBM cases and is likely a host-response metabolite. Mannose and arabinose–supposed surrogates for lipoarabinomannan, a component of the *M. tb* cell wall–were not reliable markers for *M. tb*. Further research should focus on the analysis of fatty acids in the CSF of patients with TBM.

**Funding:** The author(s) received no specific funding for this work.

**Competing interests:** The authors have declared that no competing interests exist.

## 1 Introduction

Tuberculous meningitis (TBM), the most severe extra-pulmonary presentation of tuberculosis, has a high mortality rate in young children [1]. Even when treated, late-stage TBM leaves a neurological defect in survivors [2]. TBM results from the infiltration of the blood-brain barrier by *Mycobacterium tuberculosis (M. tb)*; therefore, it affects the meninges and spinal cord [3]. Although there are no current estimates of children suffering from TBM globally, research indicates that at least 100,000 people suffer from TBM annually [3]. Huynh *et al.* [4] propose that if 2% of childhood tuberculosis is TBM, then 20,000 childhood TBM cases are estimated each year globally. Based on statistics from the World Health Organisation Global TB Report for 2020, Mason and Solomons [5] postulate that at least 80,000 children developed TBM in 2019. Although, these numbers are likely gross underestimates.

The analysis of cerebrospinal fluid (CSF) is the gold standard for the definitive diagnosis of TBM. Existing methods for diagnosing TBM all have limitations and inadequacies. Microbial diagnosis has the limitation of poor sensitivity and high turnaround time [6]. Molecular tests that involve the use of GeneXpert and Xpert Ultra tests have the challenge of giving a low negative predictive value due to the paucity of *M. tb* bacilli in the CSF of cases of TBM [6,7]. Computed tomography (CT) and magnetic resonance imaging (MRI)–brain imaging techniques, also have their own limitations, such as the high level of expertise required to interpret them and their subjectivity [8]. The measurement of an enzyme called adenosine deaminase (ADA) has been used to detect TBM; although sensitive, ADA has been found to be involved in the pathology of other infections such as bacterial meningitis and ventriculitis, making measurement of ADA an unreliable way of diagnosing TBM [9]. These points highlight the fact that we need novel approaches that provide a faster, reliable, and more efficient diagnosis of TBM. Late diagnosis, and subsequent treatment, are some of the high-risk factors responsible for high mortality and morbidity rates in TBM [4].

Traditionally, the only two CSF biochemical markers considered in the differential diagnosis of TBM are highly elevated levels of lactic acid and nearly depleted glucose levels [10]; however, these two biochemical markers are not specific to TBM. In a study carried out by Van Zyl *et al.* [11], an untargeted proton magnetic resonance ($^1$H-NMR) metabolomics approach was used to analyse the CSF of a South African paediatric cohort, and 20 CSF metabolites were identified that characterise TBM (i.e., distinguished cases of TBM from healthy control cases). This is one of many breakthroughs that have been reported using metabolomics, a scientific method that employs highly sophisticated analytical instruments and statistical analyses to define the metabolite content of biofluids such as CSF. In fact, there are several metabolomics studies that characterise the metabolic profile of CSF from TBM cases [5]. Furthermore, several studies have investigated the metabolic markers specific for *M. tb* in human biofluids. Larsson *et al.* [12], using negative-ion mass spectrometry, discovered that tuberculostearic acid (TBSA) was found in conjunction with nonadecanoic acid in the sputum of patients with confirmed tuberculosis. Numerous investigations have demonstrated the presence of mycobacterial mycolic acids in both contemporary and historical TB, pointing to the potential use of mycolic acids as reliable biomarkers of tuberculosis [13–15]. Lignoceric acid behenic acid, and hexacosanoic acid are thermal cleavage products of mycolic acids [16], while behenic acid has also been found to disintegrate into palmitic acid and stearic acid at high temperatures [17]. Hence, these degradation products of mycolic acids have the potential to act as surrogates of mycolic acids. In another study that employed GC-MS / MS, methyl nicotinate was found to be significantly different in the breath of patients with smear-positive tuberculosis versus the breath of healthy control patients [18]. Furthermore, lipoarabinomannan (LAM), a heterogeneous lipoglycan that forms part of the *M. tb* cell wall, has been found to be present in urine of

TB patients, though with low sensitivity, even with the aid of a commercial urinary LAM test kit [19,20]. D-mannose and D-arabinose, degradation products of LAM, have been reported to serve as surrogates for detection of LAM [21,22].

To the best of our knowledge, no metabolomics studies have examined the CSF from TBM cases with the aim to identify and quantify *M. tb*-specific metabolic markers. Hence, the motivation for the current study. In this study we used two analytical platforms– [1]H-NMR and two-dimensional gas chromatography coupled with time-of-flight mass spectrometry (GCxGC-TOFMS), in a targeted metabolomics investigation. The aim of this study was to identify and quantify *M. tb*-specific metabolites and their catabolic products in the CSF from TBM cases. The ultimate goal was to find *M. tb*-specific markers in the CSF of TBM cases using metabolomics techniques, in order to enhance earlier diagnosis.

## 2. Materials and methods

### 2.1 Chemicals and stock solutions

Based on the literature, ten chemical standards, identified as potential metabolites of *M. tb*, were selected for this targeted study (see Table 1).

Stock solutions of these chemical standards were prepared at a concentration of 1 mg/mL using milli-Q water for the polar standards and methanol D4 (Millipore, 811-98-3) for the non-polar standards.

Other chemicals used were: 3-(trimethylsilyl) propionic acid (TSP) (Millipore, 244923-21-8), potassium phosphate monobasic (Sigma-Aldrich, 7778-77-0), sodium phosphate monobasic (Sigma-Aldrich, 7558-80-7), methoxyamine hydrochloride (Sigma-Aldrich, 593-56-6), pyridine (Sigma-Aldrich, 110-86-1), N,O-bis(trimethylsilyl)trifluoroacetamide (BSTFA) with 1% trimethylsilylchloride (TMCS) (Sigma-Aldrich, 25561-30-2).

### 2.2 Sampling

The participants in this retrospective study (2010–2015) were children ($\leq$12 years old) from the Western Cape province of South Africa–a population endemic with tuberculosis [23]. Based on the clinical signs and symptoms of meningitis, each participant was referred from surrounding regional clinics to the tertiary-level paediatric service at the Tygerberg Academic Hospital in Cape Town [11]. Each child was evaluated by the paediatric neurology team upon arrival, and after they were stable enough, a lumbar puncture was used to obtain a CSF sample

**Table 1. Chemical standards and their details.**

| S/N | Compounds | Supplier | CAS No. | Polarity |
|-----|-----------|----------|---------|----------|
| 1. | Behenic acid[1] | Sigma-Aldrich | 112-85-6 | Non-polar |
| 2. | Hexacosanoic acid[1] | Sigma-Aldrich | 506-46-7 | Non-polar |
| 3. | Nonadecanoic acid[2] | Sigma-Aldrich | 646-30-0 | Non-polar |
| 4. | D-(+)-Mannose | Sigma-Aldrich | 3458-28-4 | Polar |
| 5. | D-(-)-Arabinose | Sigma-Aldrich | 10323-20-3 | Polar |
| 6. | Methyl nicotinic acid | Sigma-Aldrich | 93-60-7 | Polar |
| 7. | Lignoceric acid[1] | Sigma-Aldrich | 567-59-5 | Non-polar |
| 8. | α-Mycolic acid (AMA) | Avanti Polar Lipids | 23040-84-8 | Non-polar |
| 9. | Methoxy-mycolic acid (MMA) | Avanti Polar Lipids | 23599-54-4 | Non-polar |
| 10. | Keto-mycolic acid (KMA) | Avanti Polar Lipids | 2260795-20-6 | Non-polar |

1 = degradation products of mycolic acids, 2 = degradation product of tuberculostearic acid.

for standard differential diagnosis. HIV co-infection complicates an already exacerbated CSF metabolic profile; hence, the primary exclusion criterion was HIV-positive/unknown cases. For the use of the CSF samples for research, written and informed assent and/or consent were acquired. The study was approved by the Health Research Ethics Committee (HREC) of Stellenbosch University, Tygerberg Hospital (ethics approval no. N16/11/142), the Western Cape Provincial Government, as well as by the HREC of the North-West University, Potchefstroom campus (ethics approval no. NWU-00063-18-A1-04). NOTE: The authors did not have access to information that could identify individual participants during or after data collection.

Patients with bacteriologically proven TBM (definite TBM; n = 21) and non-meningitis (control; n = 25) patients comprised the two experimental groups employed in this investigation. The control group consisted of paediatric patients whose clinical symptoms suggested that they could have meningitis, but in whom special investigations excluded a diagnosis of meningitis. Additionally, we excluded control patients who had neurological symptoms such as viral encephalopathy and febrile seizures.

A uniform research case definition for TBM was used to determine a definite diagnosis of TBM [24]:

a. Clinical symptoms of meningitis, including: headache, fever, nausea, vomiting, photophobia, inflammation of the meninges, and/or neck stiffness.

b. One or more of the following:

 i. Presence of acid-fast bacilli in the CSF

 ii. CSF culture positive for *M. tb*, or

 iii. *M. tb*-positive commercial NAAT (including GeneXpert) of CSF.

Supportive investigations included a positive tuberculin skin test, a computerised tomography scan or magnetic resonance image displaying the distinctive characteristics of tuberculosis in the brain (ventricular dilatation, meningovascular enhancement, and/or granulomas), and clinical evidence of other types of extrapulmonary tuberculosis [11].

## 2.3 $^1$H-NMR spectroscopy

**2.3.1 Sample preparation.** The patient samples were previously prepared and analysed via $^1$H-NMR by our research group [11]. Briefly, all samples were fully thawed at room temperature before being prepared. A 100μL amount of CSF was centrifuged for five minutes at 12,000 g. Subsequently, the supernatant was prepared in a 2 mm NMR tube using a 180 mm long bevel-tipped needle and an eVol® NMR digital syringe. The eVol® NMR digital syringe had the following programmed pipetting sequence: (1) aspirate 6 μL of the NMR buffer solution; (2) aspirate 54 μL of the filtered sample (10%:90% ratio of $D_2O$:$H_2O$); (3) purge 60 μL (this dispenses prepared material into a 2 mm NMR tube); (4) aspirate 60 μL; (5) purge 60 μL (mix sample once within 2 mm NMR tube in order to achieve homogeneity); and the wash sequence follows: (6) aspirate 100 μL d$H_2O$; (7) purge 100 μL (waste); (8) aspirate 100 μL d$H_2O$; 9) purge 100 μL (waste); 10) aspirate 100 μL d$H_2O$; 11) purge 100 μL (waste); end.

For the pure compounds, 540 μL of the chemical compound stock solution was added to a 60 μL volume of a potassium phosphate buffer, containing TSP and either deuterium oxide (polar compounds) or methanol D4 (non-polar compounds), vortexed, and the whole volume was then transferred to a 5 mm glass NMR tube, and capped.

**2.3.2 $^1$H-NMR analysis.** Samples and standards were loaded onto a SampleXpress autosampler in a random order and were measured at 500 MHz on a Bruker Avance III HD NMR

spectrometer equipped with a 5 mm triple-resonance inverse (TXI) $^1$H {$^{15}$N, $^{13}$C} probe head and x, y, z gradient coils. The inner coil of the TXI probe was optimised for $^1$H observation, the focus of our study. The $^1$H spectra were acquired as 128 transients in 32K data points with a spectral width of 12 000 Hz and acquisition time of 2.72 s. The receiver gain was set to 64. The sample temperature was maintained at 300 K and the $H_2O$ resonance was presaturated by single-frequency irradiation (NOESY-1D) during a relaxation delay of 4 s, with a 90° excitation pulse of 8 μs. The shimming of the sample was performed automatically on the deuterium signal. Fourier transformation and phase and baseline correction were performed automatically. The quality of the spectra was verified by ensuring that the resonance line widths for the TSP and metabolites were <1 Hz. The software used for the NMR preprocessing was Bruker Topspin (V3.5) and Bruker AMIX (V3.9.14) for metabolite identification and quantification of metabolites.

## 2.4 GCxGC-TOFMS

**2.4.1 Sample preparation.** The patient samples were previously prepared and analysed by GCxGC-TOFMS by our research group (unpublished). Briefly, an internal standard stock solution containing 50 ppm of 3-phenylbutyric acid dissolved in methanol was made prior to sample preparation. The procedure for preparing the samples involved mixing 50 μL of the internal standard solution with 50 μL of each CSF sample, then drying the mixture for 45 minutes at 40°C under a light nitrogen stream. After that, each dried CSF sample received 50 μL of methoxyamine hydrochloride and it was vortexed for 30 seconds. The samples were then incubated for 90 minutes at 50°C. Lastly, 80 μL of BSTFA containing 1% TMCS was added to each sample and incubated for 60 minutes at 60°C. The derivatised samples were then placed in a 0.1 mL GC-MS insert in a glass GC-MS vial and sealed.

For this investigation, a 100 μL volume of each of the stock solutions of the ten targeted compounds was pipetted into individual glass GC-MS vials and dried under nitrogen gas on a heating block at 40°C for 45 minutes until completely dried. All samples were derivatised using 50 μL of methoxyamine hydrochloride in pyridine (15 mg/mL) at 50°C for 90 min, followed by 50 μL of BSTFA with 1% TMCS at 60°C for 60 min. The extracts were transferred to a 0.1 mL insert in a glass GC-MS vial and capped prior to GCxGC-TOFMS analysis.

**2.4.2 GCxGC-TOFMS analysis.** One microlitre of each compound extract was injected (1:5 split ratio) onto a Pegasus 4D GCxGC-TOFMS (Leco Corporation, St. Joseph, MI, USA), which comprises a two-dimensional Agilent 7890A GC (Agilent, Atlanta, GA) coupled to a time-of-flight mass spectrometer (Leco Corporation, St. Joseph, MI, USA) equipped with a Gerstel Multipurpose Sampler (Gerstel GmbH & co. KG, Eberhard-Gerstel- Platz 1, D-45473 Mülheim an der Ruhr). First-dimensional separation was achieved with a Rxi-5Sil MS primary column (29.245 m, 0.25 mm internal diameter, 0.25 μm film thickness) (Restch GmbH & Co. KG, Haan, Germany) and a Rxi-17 capillary column (1.400 m, 0.1 mm internal diameter, 0.1 μm film thickness) as the secondary column (Restch GmbH & Co. KG, Haan, Germany). The front inlet temperature was kept constant at 270°C throughout the run, ensuring rapid vaporisation. For the primary oven, the initial temperature of the GC oven was set at 70°C for 2 min, followed by an initial increase in oven temperature of 4°C/min to a final temperature of 300°C, which was maintained for 2 min. The secondary column oven temperature was set at 85°C for 2 min, then increased by 4°C/min, until a final temperature of 300°C, at which it was maintained for another 2 min. The initial modulator temperature was 100°C for 2 min, followed by a 4°C/min increase to a final temperature of 310°C held for 9 min. To control the effluent from the primary to the secondary column, cryomodulation and a hot pulse of nitrogen gas of 0.5 s were used every 3 s. The acquisition delay for each run was 450 s and the

transfer line temperature was kept constant at 270˚C, with the ion source temperature maintained at 200˚C. The detector voltage was adjusted to 1500 V with filament bias of -70 eV. Spectra were collected from 50 to 800 m/z at an acquisition rate of 200 spectra per second.

Mass spectral deconvolution, peak alignment, and peak identification were performed using Leco Corporation's ChromaTOF software (version 4.7x). Mass spectral deconvolution was performed at a signal-to-noise ratio of 100, with a minimum of three apexing peaks. To eliminate the effect of retention time shifts and create a data matrix containing the relative abundance of all compounds present in all samples, peaks with similar mass spectra and retention times were aligned using Statistical Compare, a package of ChromaTOF. Mass fragmentation patterns and their respective retention times were screened against commercially available National Institute of Standards and Technology (NIST) spectral libraries (mainlib, replib) for peak annotation, with a similarity setting of at least 80%.

**2.4.3 Creation of GCxGC-TOFMS library.** Using Leco Corporation's ChromaTOF software (version 4.7x), a library was created from the pure compounds loaded into the GCxGC-TOFMS. The retention times and masses of the targeted peaks from the pure compounds were noted. This library was then exported to the secondary computer system where the reprocessing of data occurred.

**2.4.4 Reprocessing of previously collected GCxGC-TOFMS metabolomics data using the newly created library.** All CSF samples from the previously analysed GCxGC-TOFMS experiment were imported into a newly created folder. The same data processing method was used as in the previous GCxGC-TOFMS experiment; the only difference was the inclusion of the newly created library. All CSF samples were reprocessed using this 'new' data processing method.

**2.4.5 Statistical analysis.** The preprocessing of all the data (accessed 01/02/2024) was done using Microsoft Excel. The concentration of the targeted *M. tb* metabolites in the GCxGC-TOFMS data set was calculated by normalising with the internal standard 3-phenylbutyric acid.

MetaboAnalyst (6.0) [25], a web-based server built on the statistical package "R" (version 6.0), was used to apply various univariate (unpaired t-test and box plots) and multivariate [principal components analysis (PCA) and partial least squares discriminant analysis (PLSDA)] analyses. A natural grouping or distinction between sample groups can be ascertained using PCA, an unsupervised technique. On the basis of their individual modelling capacities, the variables/metabolites that best describe this differential are rated. PLS-DA, on the other hand, is a supervised technique that finds the variables/metabolites that best describe the differentiated sample groups and is used to identify group membership of an individual sample. The relevance of each metabolite is directly correlated with its ranking based on the variables' influence on the projection (VIP) parameter [26]. For the univariate measure, a p-value < 0.05 was considered statistically significant. Biomarker analysis was also performed using MetaboAnalyst (6.0) to generate receiver operating characteristic (ROC) curves for the differentiating metabolites.

# 3. Results

## 3.1 GCxGC-TOFMS

**3.1.1 New library search.** As previously reported in the literature [16], the three mycolic acids were not visible in the GCxGC-TOFMS data because the front inlet temperature of the GCxGC-TOFMS was held constant at 270˚C, which caused the thermally labile mycolic acids to cleave into behenic acid, lignoceric acid and hexacosanoic acid. Also reported in the

LECO® ChromaTOF® optimized for Pegasus® 4D

| | Name | CAS | Library | Id |
|---|---|---|---|---|
| 1 | Lignoceric acid | 74367-37-6 | Victory | 14 |
| 2 | D-Arabinose | 18622-97-4 | Victory | 19 |
| 3 | D-Mannose | 55529-69-6 | Victory | 22 |
| 4 | Methyl nicotinate | 93-60-7 | Victory | 27 |
| 5 | Nonadecanoic acid | 74367-35-4 | Victory | 28 |
| 6 | Hexacosanoic acid | 506-46-7 | Victory | 32 |
| 7 | Behenic acid | 112-85-6 | Victory | 36 |
| 8 | Stearic acid | 18748-91-9 | Victory | 38 |
| 9 | Palmitic Acid | 55520-89-3 | Victory | 39 |

**Fig 1. GCxGC-TOFMS library of targeted metabolites for this study.**

literature [27], behenic acid cleaves into stearic acid and palmitic acid. Hence, nine metabolites (shown in Fig 1) were initially targeted for evaluation in this study.

Upon data mining the reprocessed data using the newly created library, only three of the nine metabolites could be identified, namely: D-mannose, D-arabinose, and palmitic acid.

**3.1.2 Additional search of existing NIST library.** Since only D-mannose, D-arabinose, and palmitic acid were detectable in the reprocessed data, this prompted us to expand our search to include other fatty acids because of their relationship with mycolic acids and other components of the lipid-rich *M. tb* cell wall.

The existing NIST library was used to verify the GCxGC-TOFMS CSF data for other fatty acids. Three additional fatty acids were detected; namely: propanoic acid, nonanoic acid, and undecynoic acid. Hence, six targeted metabolites were statistically evaluated in this study: D-mannose, D-arabinose, propanoic acid, nonanoic acid, undecynoic acid, and palmitic acid.

**3.1.3 Multivariate analysis.** Unsupervised principal component analysis (PCA) and supervised partial least squares discriminant analysis (PLSDA) were used to generate the scores plots in Fig 2, respectively. These scores plots qualitatively show that there is a distinction between the definite TBM group and the control group based on the six targeted metabolites evaluated in this study.

**3.1.4 Univariate analysis.** Four of the six targeted metabolites were found to be statistically significant; namely: D-mannose (FDR p-value = 0.00043), D-arabinose (FDR p-value = 0.0072), nonanoic acid (FDR p-value = 0.03) and propanoic acid (FDR p-value < 0.0001). The concentrations of these four metabolites are visually represented as box plots in Fig 3. The gas chromatographs and mass spectra patterns of these four compounds are presented S1-S4 Figs in S1 File.

**3.1.5 Biomarker analysis.** Furthermore, using the biomarker analysis function of MetaboAnalyst, Receivers Operating Characteristics (ROC) curves were generated for the four statistically significant metabolites, reported in Fig 4.

## 3.2 $^1$H-NMR

All the compounds listed in Table 1 were investigated in the patient's CSF samples. Of these 10 compounds, only D-mannose and D-arabinose were barely detectable via analysis of the 1D $^1$H-NMR spectra–see S5 Fig in S1 File. The presence of D-mannose and D-arabinose was confirmed by 2D (COSY) NMR analysis. Therefore, the $^1$H-NMR data were able to confirm the identity of D-mannose and D-arabinose from the GCxGC-TOFMS data. Although the identity

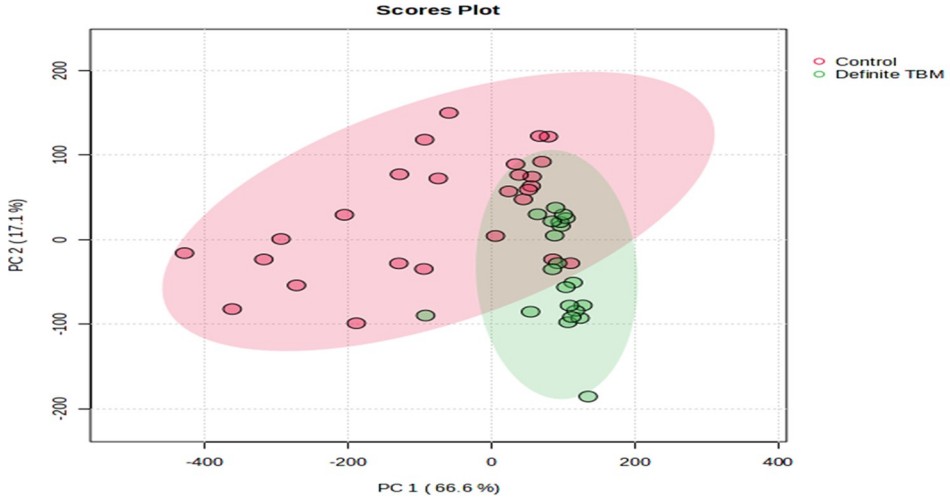

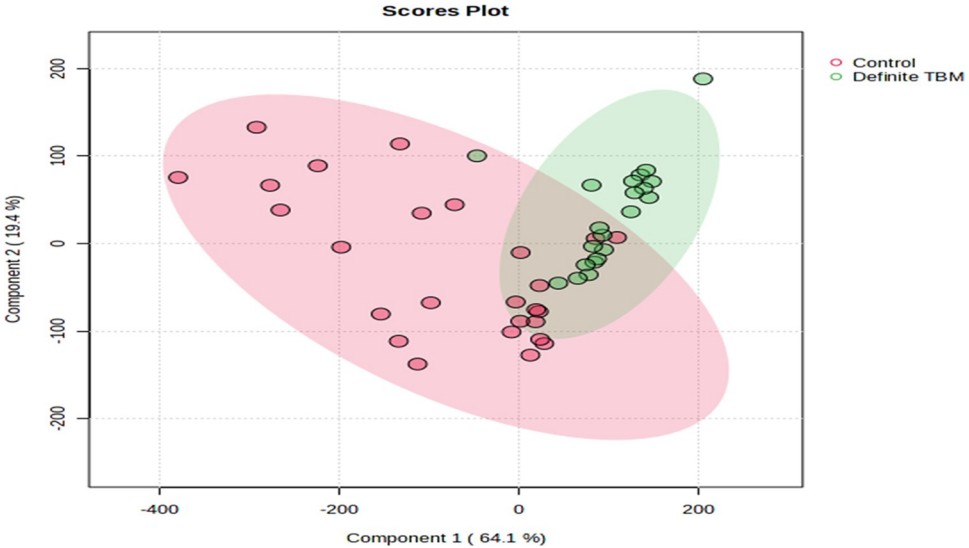

**Fig 2.** PCA (top) and PLSDA (bottom) scores plots for the six targeted metabolites: D-mannose, D-arabinose, propanoic acid, nonanoic acid, undecynoic acid, and palmitic acid. Both plots show some, but not complete, separation between the Def TBM [28] and control (red) groups.

of D-mannose and D-arabinose could be confirmed, these two metabolites could not be reliably quantified using the [1]H-NMR data, due to their very low levels of concertation (close to noise level). Herein lies the inherent challenge with [1]H-NMR–it is not as sensitive as its MS counterpart; however, its complementary nature with GCxGC-TOFMS allowed us to confirm the identity of D-mannose and D-arabinose.

## 4. Discussion

Targeted metabolites (Table 1) known to be linked to *M. tb* (i.e., markers of *M. tb*) were investigated in the CSF of definite TBM cases. Two sugars–D-mannose and D-arabinose, were identified in the GCxGC-TOFMS data and found to be statistically significant. Three *M. tb* compounds [α-mycolic acid (C80), keto-mycolic acid (C86), and methoxy-mycolic acids

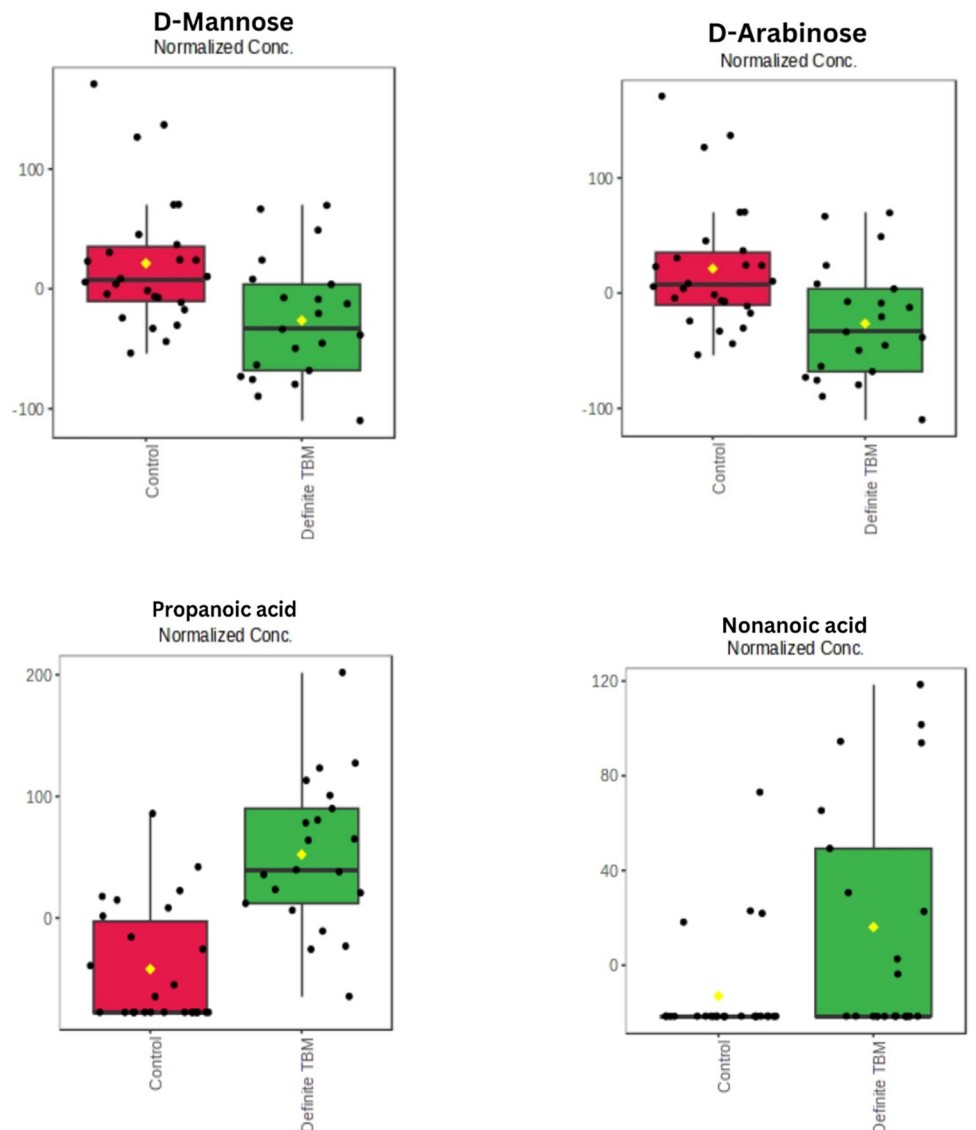

**Fig 3. Box plots of the four statistically significant targeted metabolites.** Concentrations (x-axis) are in micromol per L.

(C85)] cleaved upon injection into the GCxGC-TOFMS and were not detectable. None of the expected degradation products of mycolic acids [hexacosanoic acid (C26:0), lignoceric acid (C24:0), behenic acid (C22:0), stearic acid (C18:0), and palmitic acid (C16:0)], nor the degradation product of tuberculostearic acid–nonadecanoic acid, were detectable in the GCxGC-TOFMS data. The inability to detect known degradation products of the unique lipids of the cell wall of *M. tb* are consistent with existing diagnostic challenges; in particular, the low negative predictive value of molecular testing (GeneXpert and Xpert Ultra). The underlying reason is probably the same here–there is a paucity of *M. tb* bacilli in the CSF of TBM cases, especially in the early stages of TBM. Hence, the inability to detect direct markers of *M. tb*.

Additional fatty acids, as potential further downstream catabolites, were investigated and some (propanoic acid, nonanoic acid, and undecynoic acid) were identified in the GCxGC-TOFMS CSF data. Ultimately, only six metabolites were identified and four

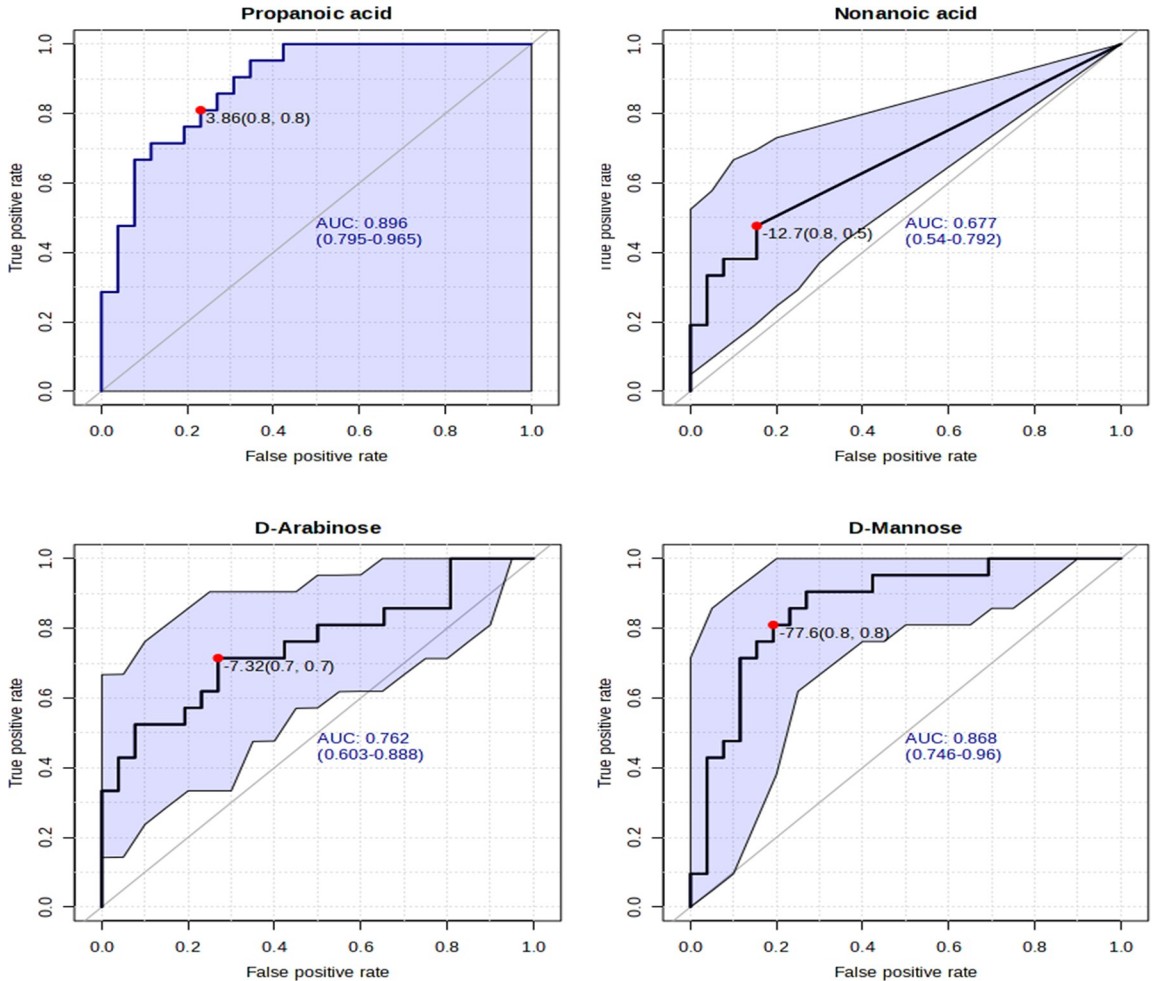

**Fig 4. ROC curves for the four statistically significant metabolites in this study.** Propanoic acid showed the best potential as a biomarker for TBM.

(propanoic acid, nonanoic acid, D-mannose, and D-arabinose) of these six metabolites were statistically significant in differentiating TBM from controls. These four metabolites are discussed further.

## 4.1 Propanoic (propionic) acid

Propanoic acid was found in higher concentration in the definite TBM samples, and it had an AUC of 0.896 in a ROC curve analysis, making it the best potential marker for TBM. Propanoic acid is a short-chain fatty acid that can be used by the host to produce glucose in the liver through gluconeogenesis to account for the increased need for glucose for perturbed neuroenergetics in the brain due to the invasion of *M. tb* and the innate immune response [29]. Propanoic acid has also been reported to possess anti-inflammatory properties [30], and may be part of the host's neuroinflammatory response associated with TBM [31].

In the gut, propanoic acid is produced by microbiota during the fermentation of dietary fibres [32]. Bidirectional neuronal, endocrine, and immunological connections connect the gut and brain to form the gut-brain axis [33]. Disorders in the composition and amount of gut bacteria can impact both the enteric and central nervous systems (CNS), implying the presence

of a microbiota-gut-brain axis [34–36]. 3-Hydroxypropionic acid (3-HPA)–an intermediary in the degradation process of gut-produced propionic acid–was also found in elevated concentrations in the urine in all of the three stages of paediatric TBM undergoing treatment [37].

Moreover, Savvi [38] showed that in the absence of a functional methylcitrate cycle and with inhibition of the glyoxylate cycle, *M. tb* revert to using propionic acid as a source of carbon for energy production. Therefore, propanoic acid is a ubiquitous metabolite and its source could be either from the host, gut microbiota or directly from the *M. tb*.

## 4.2 Nonanoic acid

Nonanoic acid (C9:0; also known as pelargonic acid) was significantly elevated in the TBM cases in this study; however, it had the worst ROC curve of the four evaluated metabolites. Quantifiable levels of nonanoic acid were found in only four of the 25 control cases (16%), while 11 of the 21 TBM cases (52%) had nonanoic acid. The complete absence of nonanoic acid in some of the cases is probably the reason for the poor performance of nonanoic acid as a marker. It is recommended that a sample preparation method that extracts fatty acids should be used in future studies to assess this compound.

Nonanoic acid–a straight-chain saturated fatty acid, has been reported to possess antimicrobial properties [39,40]. A possible reason for the elevated concentration of nonanoic acid in the definite TBM group as compared to the control group could be that the body produced nonanoic acid as a form of resistance to *M. tb*; however, most, if not all, of the investigated paediatric TBM cases have been diagnosed with a *M. tb* infection for the first time, making it unlikely that they have developed any form of resistance to *M. tb*., yet.

To the best of our knowledge, this is the first time nonanoic acid has been reported in a study related to TBM and *M. tb*. Since nonanoic acid is an exogenous metabolite, this C9:0 fatty acid may be a unique downstream catabolite of one of the cell wall components of *M. tb*. Further research needs to be carried out to elucidate whether nonanoic acid is linked to *M. tb*.

## 4.3 Mannose and arabinose

In this study, the concentrations of mannose and arabinose were lower in the definite cases of TBM compared with the control group. This is contrary to the expected result as seen in the literature, where mannose and arabinose have been used as surrogates for the presence of LAM [41]. A possible explanation for the decreased levels of these two sugars in the TBM cases is that the host utilizes these carbohydrate sources during the metabolic burst that has been described to occur in TBM [11]. Furthermore, specific GC-MS chemical derivatization protocols for D-mannose and D-arabinose, described in the literature [21], were not carried out in this study. These protocols would have enhanced the sensitivity and specificity of the GCxGC-TOFMS to these sugars.

[1]H-NMR analysis, also used in this study, was able to detect mannose and arabinose–confirming the identities of these two sugars; however, because of their very low concentrations, they were not quantifiable (i.e., too close to the noise) in this study. An NMR spectrometer of greater strength (e.g., a 700 MHz NMR with a cryoprobe) would likely be able to quantify D-mannose and D-arabinose in the CSF of TBM cases.

## 5. Conclusion

This study makes the first mention of nonanoic acid in the CSF of patients with definite TBM. Nonanoic acid does not appear to be an endogenous metabolite and was found to be significantly elevated in the TBM cases. Hence, we postulate that nonanoic acid is a downstream degradation product of the cell wall component of *M. tb*, this makes it a potential *M. tb*-specific

marker of TBM. More studies are needed to determine whether the cell wall components of *M. tb* degrade to form nonanoic acid. Propanoic acid was also significantly elevated in the TBM group in this study. Propanoic acid has a definite role in neuro-energetics and anti-inflammation in the host, which may explain its abundance in TBM samples, and is a source of carbon for the *M. tb*. However, more research is needed to confirm whether propionic acid is linked specifically to *M. tb*. Mannose and arabinose, the two sugars identified in this study as being statistically significant, did not produce the results expected–significantly decreased in the TBM group. This is likely because these sugars were used as energy sources for the metabolic burst known to occur in TBM. Lastly, other fatty acids such as palmitic and stearic acid were expected to be found in this cohort, but only palmitic acid was found and it was quite insignificant compared to the result of the data analysis. A larger prospective cohort study will need to test the sensitivity of the four metabolites identified in this study. Also, an addition of other types of meningitis and other communicable and non-communicable diseases of the CNS would be needed to verify the robustness of these four metabolites. After validating these results then the clinical implications/applications of these metabolites in TBM can be discussed and the rationale for their use as biomarkers of TBM can be made.

The inability to detect some of the expected metabolites that are linked to *M. tb* may be due to inadequate sample preparation; therefore, subsequent studies should focus on fatty acid analysis. Moreover, the paucity of *M. tb* bacilli in the CSF of TBM cases, especially in the early stages of TBM, is a well-known problem that also affects the molecular diagnosis of TBM–an unresolved challenge that needs to be addressed.

## Supporting information

**S1 Data.**
(XLSX)

**S1 File.**
(DOCX)

## Author Contributions

**Conceptualization:** Shayne Mason.

**Data curation:** Victory Samuel, Regan Solomons.

**Formal analysis:** Victory Samuel.

**Investigation:** Victory Samuel, Regan Solomons.

**Methodology:** Shayne Mason.

**Project administration:** Shayne Mason.

**Resources:** Regan Solomons, Shayne Mason.

**Supervision:** Shayne Mason.

**Writing – original draft:** Victory Samuel.

**Writing – review & editing:** Regan Solomons, Shayne Mason.

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
