## [Decision Letter · Decision Letter 0]

15 Sep 2024

PONE-D-24-32616Targeted metabolomics investigation of metabolic markers of Mycobacterium tuberculosis in the cerebrospinal fluid of paediatric patients with tuberculous meningitisPLOS ONE

Dear Dr. Mason,

Thank you for submitting your manuscript to PLOS ONE. After careful consideration, we feel that it has merit but does not fully meet PLOS ONE’s publication criteria as it currently stands. Therefore, we invite you to submit a revised version of the manuscript that addresses the points raised during the review process.

We look forward to receiving your revised manuscript.

Kind regards,

Guocan Yu

Academic Editor

PLOS ONE

Journal Requirements:

Reviewers' comments:

Reviewer's Responses to Questions

**Comments to the Author**

1. Is the manuscript technically sound, and do the data support the conclusions?

Reviewer #1: Yes

Reviewer #2: Yes

Reviewer #3: Yes

2. Has the statistical analysis been performed appropriately and rigorously? 

Reviewer #1: Yes

Reviewer #2: Yes

Reviewer #3: Yes

3. Have the authors made all data underlying the findings in their manuscript fully available?

Reviewer #1: Yes

Reviewer #2: Yes

Reviewer #3: Yes

4. Is the manuscript presented in an intelligible fashion and written in standard English?

Reviewer #1: Yes

Reviewer #2: Yes

Reviewer #3: Yes

5. Review Comments to the Author

Reviewer #1: appreciate the inquisitiveness of the authors in searching for MTB-specific biomarkers in CSF. However, I am curious about the rationale behind selecting a control group for this study. While I understand that a control group can help determine what is specific to MTB, this specificity seems to be evident from the results. Could the authors clarify the necessity and role of the control group in this context?

Reviewer #2: Title: Consider emphasizing the novelty or impact of the study in the title. For instance, adding a phrase like "A Novel Approach to Biomarker Discovery" could enhance its appeal.

Abstract: Additionally, mentioning the potential clinical applications in the abstract would strengthen its relevance.

Study objective: Did you consider framing the objective in a hypothesis-driven manner, which could make it more focused and testable? Given the hurdles in pediatric TB diagnosis and treatment.

Study design: Retrospective - not sure whether you sifted through consent forms where approval was granted for future studies or you consented afresh (it is not clear when reading - make it concise how you accessed CSF samples from 12 year olds and under). Explain how potential confounding factors (e.g., co-infections, treatment history) were controlled or accounted for.

Data analysis: Why was bivariate analysis skipped? Was it considered by the study team even with small sample size?

Results: Consider reorganizing the results to flow more logically from general findings to specific metabolic markers. This will help guide the reader through the data more smoothly. For example you have started with multivariate then univariate analysis yet it should be the other way round. Clearly distinguish between statistically significant findings and trends that may require further investigation.

Discussion: The potential for these markers to be used in clinical settings, such as in diagnostic tests or monitoring disease progression. Providing a practical application of the findings will enhance the study's impact. Discuss the need for validation in larger, more diverse prospective cohorts. Additionally, explore how these findings compare to similar studies in adult populations, if applicable.

Conclusion: Consider providing more specific suggestions for future studies, such as longitudinal studies to assess the temporal dynamics of these metabolic markers or exploring their relevance in other forms of tuberculosis. Reiterate the study’s potential clinical impact more strongly, emphasizing how it could contribute to earlier diagnosis or better treatment strategies for pediatric patients with tuberculous meningitis.

Final thoughts: This manuscript makes a meaningful contribution to the field of pediatric infectious diseases and metabolomics. With minor adjustments, as flagged for the attention of the authors the manuscript can be significantly strengthened. These findings have the potential to pave the way for future research and clinical applications, making it a valuable addition to the literature on tuberculous meningitis.

Reviewer #3: In this manuscript, Victory Samuel and Shayne Mason explore novel TBM diagnosis biomarker using targeted metabolomics analysis of CSF sample from patients with TBM and non-meningitis. They included a 6-targeted metabolites list (a 6x46 dataframe) to perform PCA and PLSDA analysis, T test, and ROC analysis. The authors claimed that nonanoic acid’s existence in the CSF indicated its specifically derived from Mtb, and their short of detection of expected metabolites highlighted the adequate sample preparation. Overall, this manuscript provides a moderate targeted metabolites analysis dataset of the CSF samples. The manuscript could be improved in a few aspects, which are listed below:

1. The analysis largely depends on MetaboAnalyst, which requires a pertinent citation.

2. It is important to mention all the supplementary figures in appropriated sections in the manuscript.

3. The figure legends are too brevity to provide enough information for the readers. Please revise these legends.

4. For the unpaired t test used in Fig.3, the FDR adjusted p value is q value, in fact. Please correct the section 2.4.5, “a p-value <0.05 was” should be “a q-value <0.05”.

5. For the ROC analysis, the authors need comment their analysis results, such as whether the AUC values indicate a rationale for biomarker kit development? And the authors could collectively perform 2,3,4 markers-combined ROC analysis to discriminate the TBM and control group.

6. See page 14 Section 3.2, is it Table 1 or Figure 1, and how many compounds, 9 or 10?

7. The biology of the 4 statically significant metabolites seems beyond the aim of this study, as the aim t is discover novel biomarkers from CSF samples. The reviewer suggested a simple summary is enough.

6. PLOS authors have the option to publish the peer review history of their article (what does this mean?). If published, this will include your full peer review and any attached files.

Reviewer #1: No

Reviewer #2: **Yes: **Dr. Steve Wandiga

Reviewer #3: **Yes: **Rui Yang

---

## [Author Response · Author response to Decision Letter 0]

7 Nov 2024

Reviewer #1: appreciate the inquisitiveness of the authors in searching for MTB-specific biomarkers in CSF. However, I am curious about the rationale behind selecting a control group for this study. While I understand that a control group can help determine what is specific to MTB, this specificity seems to be evident from the results. Could the authors clarify the necessity and role of the control group in this context?

Response: Metabolomics studies (untargeted and targeted), especially human/health based metabolomics studies, are typically case-control (experimental group vs control group) type studies. As the reviewer states, the necessity of a control group is to help determine what is specific to the experimental group (TBM). The control group is also a necessity for the multivariate statistical analysis (PCA and PLSDA) to visualize [qualitatively] the differentiation of the experimental group (TBM) from the healthy controls, and to identify the variables (metabolites) that are responsible for differentiating CSF from TBM cases from ‘normal’ values.

Reviewer #2: Title: Consider emphasizing the novelty or impact of the study in the title. For instance, adding a phrase like "A Novel Approach to Biomarker Discovery" could enhance its appeal.

Response: Thank you for this suggestion. However, the use of metabolomics as the scientific method used in this study is not a novel approach.

Abstract: Additionally, mentioning the potential clinical applications in the abstract would strengthen its relevance.

Response: We have not yet validated our findings. A larger prospective cohort study will need to test the sensitivity and specificity of the 4 metabolites identified in this study. Also, an addition of other types of meningitis and other communicable and non-communicable diseases of the CNS would be needed to verify the robustness of these metabolites. Only thereafter would we be able to discuss the clinical implications/applications of these metabolites.

We have added this to the manuscript.

Study objective: Did you consider framing the objective in a hypothesis-driven manner, which could make it more focused and testable? Given the hurdles in pediatric TB diagnosis and treatment.

Response: This is a targeted metabolomics study (as described in the title) and, by design, it is aimed at testing a hypothesis. The hypothesis of this study is that there are metabolites specific to M.tb in the CSF of TBM cases. This is articulated in the paper as our aim: “identify and quantify M. tb-specific metabolites and their catabolic products in the CSF from TBM cases”, with the broader aim ”to find M. tb-specific markers in the CSF of TBM cases using metabolomics techniques, in order to aid in earlier diagnosis.” 

Study design: Retrospective - not sure whether you sifted through consent forms where approval was granted for future studies or you consented afresh (it is not clear when reading - make it concise how you accessed CSF samples from 12 year olds and under). Explain how potential confounding factors (e.g., co-infections, treatment history) were controlled or accounted for.

Response: All considerations of regarding ethics were addressed by two Ethics Committees – NWU and SUN (see pg 6 and the ethical statement in the paper). The TBM group had no infections other than M.tb at the point of sample collection. And, as described in a previous response, a larger prospective cohort is needed to evaluate potential confounding factors.

Data analysis: Why was bivariate analysis skipped? Was it considered by the study team even with small sample size?

Response: Bivariate analysis and correlation analyses were done but nothing useful came from the results and were not included in this paper.

Results: Consider reorganizing the results to flow more logically from general findings to specific metabolic markers. This will help guide the reader through the data more smoothly. For example, you have started with multivariate then univariate analysis, yet it should be the other way round. Clearly distinguish between statistically significant findings and trends that may require further investigation.

Response: The typical metabolomics statistical workflow starts with multivariate analyses first - first unsupervised (PCA), then supervised (PLSDA). This is because these multivariate statistical analyses are often more robust and are used to identify which variables are of interest; essentially, reducing the number of variables before doing univariate statistical analyses. In larger metabolomics data sets it is impractical to do univariate statistical analyses on every variable. For example, if the data set contains 1000+ metabolites, but only 4 are responsible for differentiating between two groups, then it is easier to do the multivariate statistical analyses first to identify these 4 metabolites for more in-depth univariate analyses. Additionally, performing univariate statistical analyses before PCA removes the strength of the unsupervised approach of PCA.

Discussion: The potential for these markers to be used in clinical settings, such as in diagnostic tests or monitoring disease progression. Providing a practical application of the findings will enhance the study's impact. Discuss the need for validation in larger, more diverse prospective cohorts. Additionally, explore how these findings compare to similar studies in adult populations, if applicable.

Response: Thank you for this suggestion. We have added that a larger prospective cohort is needed to validate our results. We cannot make practical/clinical suggestions of the 4 metabolites identified in this study until the results have been validated. There are no other metabolomics studies (pediatric or adult) aimed specifically at M.tb metabolites to compare our results against.

Conclusion: Consider providing more specific suggestions for future studies, such as longitudinal studies to assess the temporal dynamics of these metabolic markers or exploring their relevance in other forms of tuberculosis. Reiterate the study’s potential clinical impact more strongly, emphasizing how it could contribute to earlier diagnosis or better treatment strategies for paediatric patients with tuberculous meningitis.

Response: We have added to the conclusions.

Final thoughts: This manuscript makes a meaningful contribution to the field of paediatric infectious diseases and metabolomics. With minor adjustments, as flagged for the attention of the authors the manuscript can be significantly strengthened. These findings have the potential to pave the way for future research and clinical applications, making it a valuable addition to the literature on tuberculous meningitis.

Response: Thank you to the Reviewer for the positive comments.

Reviewer #3: In this manuscript, Victory Samuel and Shayne Mason explore novel TBM diagnosis biomarker using targeted metabolomics analysis of CSF sample from patients with TBM and non-meningitis. They included a 6-targeted metabolites list (a 6x46 dataframe) to perform PCA and PLSDA analysis, T test, and ROC analysis. The authors claimed that nonanoic acid’s existence in the CSF indicated its specifically derived from M. tb, and their short of detection of expected metabolites highlighted the adequate sample preparation. Overall, this manuscript provides a moderate targeted metabolites analysis dataset of the CSF samples. The manuscript could be improved in a few aspects, which are listed below:

1. The analysis largely depends on MetaboAnalyst, which requires a pertinent citation.

Response: A citation for MetaboAnalyst has been added to page 10 of the manuscript.

2. It is important to mention all the supplementary figures in appropriated sections in the manuscript.

Response: The following has been added on pg 13: The gas chromatographs and mass spectra patterns of these four compounds are presented in the supplementary information as Figures S1 -S4. 

The NMR spectra of mannose and arabinose as Figure S5 in the supplementary is already described in manuscript on pages 14-15.

3. The figure legends are too brevity to provide enough information for the readers. Please revise these legends.

Response: Additional information has been added to the figure legends.

4. For the unpaired t test used in Fig.3, the FDR adjusted p value is q value, in fact. Please correct the section 2.4.5, “a p-value <0.05 was” should be “a q-value <0.05”.

Response: Figure 3 shows the box plots of the concentrations of the significant metabolites and does not include p-values or q-values. The p-value in 2.4.5 is the correct output.

5. For the ROC analysis, the authors need comment their analysis results, such as whether the AUC values indicate a rationale for biomarker kit development? And the authors could collectively perform 2,3,4 markers-combined ROC analysis to discriminate the TBM and control group.

Response: Combined ROC analyses were performed but the results were not statistically significant, hence they were not reported. We have added the following: A larger prospective cohort study will need to test the sensitivity of the four metabolites identified in this study. Also, an addition of other types of meningitis and other communicable and non-communicable diseases of the CNS would be needed to verify the robustness of these four metabolites. After validating these results then the clinical implications/applications of these metabolites in TBM can be discussed and the rationale for their use as biomarkers of TBM can be made

6. See page 14 Section 3.2, is it Table 1 or Figure 1, and how many compounds, 9 or 10?

Response: Section 3.2 correctly refers to Table 1 indicating the original 10 compounds of interest. Figure 1 shows nine compounds. As described in the text, the three mycolic acids (from Table 1) were cleaved, due to the high thermal conditions, to stearic acid and palmitic acid. Hence, from 10 compounds in Table 1 to 9 compounds in Figure 1.

7. The biology of the 4 statically significant metabolites seems beyond the aim of this study, as the aim is to discover novel biomarkers from CSF samples. The reviewer suggested a simple summary is enough.

Response: We believe that the biological interpretation in the discussion gives context to these 4 metabolites and how they potentially relate to M.tb. The discussion describes how nonanoic acid is exogenous, that mannose and arabinose are not increased in TBM, as originally expected, but are decreased and the reason why we think this is so, and how propanoic acid is ubiquitous. This biological context improves the citeability of the paper because it gives potential directives for follow-up studies. However, as advised by the Reviewer, we have reduced the length of the discussion.

---

## [Decision Letter · Decision Letter 1]

19 Nov 2024

Targeted metabolomics investigation of metabolic markers of Mycobacterium tuberculosis in the cerebrospinal fluid of paediatric patients with tuberculous meningitis

PONE-D-24-32616R1

Dear Dr. Mason,

We’re pleased to inform you that your manuscript has been judged scientifically suitable for publication and will be formally accepted for publication once it meets all outstanding technical requirements.

Kind regards,

Guocan Yu

Academic Editor

PLOS ONE

Additional Editor Comments (optional):

Reviewers' comments:

Reviewer's Responses to Questions

**Comments to the Author**

1. If the authors have adequately addressed your comments raised in a previous round of review and you feel that this manuscript is now acceptable for publication, you may indicate that here to bypass the “Comments to the Author” section, enter your conflict of interest statement in the “Confidential to Editor” section, and submit your "Accept" recommendation.

Reviewer #2: All comments have been addressed

Reviewer #3: All comments have been addressed

2. Is the manuscript technically sound, and do the data support the conclusions?

Reviewer #2: Yes

Reviewer #3: Yes

3. Has the statistical analysis been performed appropriately and rigorously? 

Reviewer #2: Yes

Reviewer #3: Yes

4. Have the authors made all data underlying the findings in their manuscript fully available?

Reviewer #2: Yes

Reviewer #3: Yes

5. Is the manuscript presented in an intelligible fashion and written in standard English?

Reviewer #2: Yes

Reviewer #3: Yes

6. Review Comments to the Author

Reviewer #2: Adjustments are appropriate and for elements that were not possible in this work your team can consider those comments in the next paper

Reviewer #3: This new version provides some updates and fixes some misleading mentioned before. However, I recommend the author that "FDR adjusted p value" is "q value". Thus, " For the univariate measure, a p-value < 0.05

was considered statistically significant" , the "p-value" in fact is "FDR p-value", which also called "q value". I think, " For the univariate measure, a FDR p-value < 0.05 (or q value <0.5) was considered statistically significant" is the right one.

7. PLOS authors have the option to publish the peer review history of their article (what does this mean?). If published, this will include your full peer review and any attached files.

Reviewer #2: **Yes: **Steve Wandiga

Reviewer #3: **Yes: **Rui Yang

---

## [Editor Report · Acceptance letter]

6 Dec 2024

PONE-D-24-32616R1 

PLOS ONE

Dear Dr. Mason, 

I'm pleased to inform you that your manuscript has been deemed suitable for publication in PLOS ONE. Congratulations! Your manuscript is now being handed over to our production team.

Kind regards, 

on behalf of

Dr. Guocan Yu 

Academic Editor

PLOS ONE